# Anti-TBE Intrathecal Synthesis as a Prediction Marker in TBE Patients

**DOI:** 10.3390/pathogens11040416

**Published:** 2022-03-29

**Authors:** Agnieszka Siemieniako-Werszko, Piotr Czupryna, Anna Moniuszko-Malinowska, Justyna Dunaj-Małyszko, Sławomir Pancewicz, Sambor Grygorczuk, Joanna Zajkowska

**Affiliations:** Department of Infectious Diseases and Neuroinfections, Medical University of Białystok, Żurawia 14, 15-540 Białystok, Poland; neuroin@umb.edu.pl (A.S.-W.); anna.moniuszko-malinowska@umb.edu.pl (A.M.-M.); justyna.dunaj-malyszko@umb.edu.pl (J.D.-M.); slawomir.pancewicz@umb.edu.pl (S.P.); sambor.grygorczuk@umb.edu.pl (S.G.); joanna.zajkowska@umb.edu.pl (J.Z.)

**Keywords:** TBE, intrathecal synthesis, sequelae

## Abstract

Tick-borne encephalitis (TBE) is an emerging vector-borne disease in Europe caused by tick-borne encephalitis virus (TBEV), which belongs to Flaviviridae. Although most of the patients quickly recover from TBE, some require further neurological and psychiatric treatment due to persistent symptoms. The aim of the study was to evaluate the usefulness of an antibodies index for predicting the course of the disease and potential persistent sequalae. Sixty-six patients (49 males and 17 females, mean age 45.97 ± 13.69 years) with TBE hospitalized in the Department of Infectious Diseases and Neuroinfections, Medical University of Bialystok, Poland, in years 2016–2019 were included to the study. TBE antibodies titer in serum and CSF samples were measured with an Anti-TBEV ELISA (IgM, IgG) EUROIMMUN test. Patients who developed persistent sequelae after TBE had significantly lower IgG intrathecal index at admission. Additionally, IgG2/IgG1was significantly higher in patients who developed sequelae. IgG intrathecal index might be a useful tool for the prediction of TBE sequelae development.

## 1. Introduction

Tick-borne encephalitis (TBE) is an emerging vector-borne disease in Europe caused by tick-borne encephalitis virus (TBEV), which belongs to *Flaviviridae*. There are three subtypes of TBEV: European (endemic in rural and forested areas of central, eastern and northern Europe), Far-eastern (endemic in far-eastern Russia and in forested regions of China and Japan), Siberian subtype (endemic in Urals region, Siberia and far-eastern Russia, and also in some areas in north-eastern Europe) [1]. The number of TBE cases in Europe varies depending on country and ranges from a few cases a year up to 1000 cases per year [2]. In Poland 200–300 TBE cases are reported annually. The cases from Podlaskie voivodship, where the Department of Infectious Diseases and Neuroinfections is located, consist of ca. 46% of all cases noticed in our country [3]. In Poland, TBE is caused by the European subtype of TBEV. The TBE incidence in Poland in 2020 was 0.41/100,000 inhabitants, while in Podlaskie region it was 6.63/ 100,000 inhabitants [3]. 

TBE may take the clinical course of meningitis, meningoencephalitis or meningoencephalomyelitis. The disease is usually mild with mortality rate ca. 2% [4].

Although most of the patients quickly recover from the TBE, some require further neurological and psychiatric treatment due to persistent symptoms.

So far, the diagnosis has been based on the presence of specific anti-TBEV IgM and IgG antibodies in serum and CSF. This method has its own limitations. TBE antibodies appear 0–6 days after onset and are usually detected when neurological symptoms are present. Specific IgM antibodies can persist for up to 10 months in vaccines or individuals who acquired the infection naturally; IgG antibody cross-reaction is possibly observed with other flaviviruses. Detection by PCR methods also has limitations, as the virus is present on the CSF for very short period of time before clinical manifestation. 

Antibody indexes are not novel methods and require specific expertise. Their niche has expanded substantially in recent years due to increasing evidence of their performance in diagnosing viral infections [5].

The aim of the study was to evaluate the usefulness of an antibodies index for predicting the course of the disease and potential persistent sequelae. 

## 2. Results

### 2.1. General Results

The comparison of patients with monophasic and biphasic course of the disease showed that the IgG intrathecal index at admission was significantly higher in patients with a monophasic course of the disease, while in the control examination, the indexes did not differ significantly. Additionally, the IgG2/IgG1 ratio was higher in patients with a biphasic course (Table 1). 

No statistical difference was stated when comparing the IgG intrathecal index between clinical course (meningitis vs. encephalitis) and sex. 

Although the comparison of the IgG intrathecal index at admission and in control lumbar punction between patients with or without co-infection with Borrelia burgdorferi did not show any significant differences, the IgG2/IgG1 ratio was significantly lower in patients with co-infection. 

Patients who developed persistent sequelae after TBE had significantly lower IgG intrathecal index at admission. Moreover, the IgG2/IgG1was significantly higher in patients who developed sequelae. 

### 2.2. ROC Analysis

ROC curve analysis indicates that IgG1 intrathecal index differentiated sequelae from non-sequelae group TBE patients. At the cut-off at 2.5984, specificity was 62.3%, and sensitivity was 67.9%. AUC  =  0.654, *p* < 0.05 (Figure 1). 

ROC curve analysis indicates that IgGII/IgGI ratio differentiated sequelae from non-sequelae group TBE patients. At the cut-off at 2.2478, specificity was 71.4%, and sensitivity was 59.3%. AUC  =  0.689, *p* < 0.05 (Figure 2).

### 2.3. Analysis of Correlation

We observed negative correlation of IgG I with duration of the first phase (r = −0.39 *p* < 0.05), pleocytosis in first sample (r = −0.29 *p* < 0.05), albumin concentration in first sample (r = −0.32, *p* < 0.05).

## 3. Discussion

Although TBE usually follows a mild course, it might lead to the development of various sequelae. So-called post-TBE syndrome may be present even in 58% of TBE patients. It may result in long-term morbidity that often affects patient’s quality of life and forces a change in lifestyle and most frequently includes cognitive or neuropsychiatric complaints (reduced stress tolerance and impaired ability to memorize), balance disorders, headache, dysphasia, hearing defects and limb paresis [6].

Our previous study, conducted on over 1000 patients from the same location as the current study, indicated that 1 month after discharge from hospital, 20.6% of patients with TBE suffered from various sequelae—usually mild, subjective symptoms such as headache or vertigo. In some patients, these symptoms were present even 10 years after the acute phase of the disease [6]. 

In the current study, we observed that ca. 47% of patients suffered from sequelae (this result should be interpreted with care, as it might by biased by the small number of patients). The dominating symptoms were headache and fatigue.

So far, the exact pathogenesis of post-TBE syndrome is not fully known. Prediction of such complications might greatly influence the long-term care of patients with a history of TBE.

Our study shows that the TBE antibody index may be a useful tool in prediction of the development of sequelae. Both IgG intrathecal index counted at admission as well as the ratio calculated by division of the control IgG intrathecal index and initial IgG intrathecal index differentiated patients who recovered completely from those who developed sequelae. 

Our observations are in line with other studies. 

Shamier et al. observed that antibody indexes can support the diagnosis of a spectrum of viral infections in immune privileged sites such as the central nervous system and the eye, through the demonstration of virus-specific intrathecal or intraocular antibody production. This is especially useful in situations where PCR has a lower positivity rate: infections with rapid viral clearance due to natural immunity or treatment and chronic stages of viral infections [5].

Radzisauskiene et al. reported that delayed IgG intrathecal synthesis in TBE is associated with a more severe course of the disease. In our study, the intrathecal synthesis did not correlate directly with clinical course of the disease; however, patients, who developed persistent sequelae had significantly lower IgG index at admission [7].

In our study, negative correlations of the IgG intrathecal index with pleocytosis and albumin concentration in CSF were observed. This indicates that early serological response has protective features and diminishes the inflammatory response in the CSF.

IgG intrathecal index values around 1 indicate equal AH and serum antibody ratios—thus, the absence of intraocular synthesis of pathogen-specific antibodies. Although there is limited literature on the ideal cut-off, there is a consensus that a cut-off of 3 provides the best trade-off between sensitivity and specificity [8,9]. The IgG intrathecal index is less reliable when there is pronounced dysfunction of the blood–ocular barrier, and an alternative calculation including a correction was proposed by Quentin [10].

It is worth mentioning that intrathecal synthesis is a routine diagnostic method in neuroborreliosis confirmation [11]. Our study adds information regarding anti-TBE intrathecal synthesis as a prediction marker in tick-borne disease other than borreliosis. 

A limitations of the study is the small number of patients examined. Additionally, there were no patients with meningoencephalomyelitis included, which may have influenced on the results. The study is based only on the IgG index, although further studies with detection of both IgM and IgG synthesis are planned.

## 4. Materials and Methods

Participants were 66 patients (49 males and 17 females, mean age 45.97 ± 13.69 years) with TBE. 

I. Clinical Criteria:

Patients hospitalized in the Department of Infectious Diseases and Neuroinfections, Medical University of Bialystok, Poland, in years 2016–2019 were included to the study. All patients had follow-up visits 1 month after the first hospitalization. 

Medical data, such as patients age, sex, symptoms and sequelae were analyzed (Table 2). Laboratory parameters such as data from cerebrospinal fluid (CSF) and serum are presented in Table 3. None of the patients were vaccinated against TBE.

The patients enrolled in the study fulfilled clinical, laboratory and epidemiological criteria of the TBE definition [12]: 

Any person with symptoms of inflammation of the CNS (e.g., meningitis, meningoencephalitis, encephalomyelitis, encephaloradiculitis).

II. Laboratory Criteria: 

—Laboratory criteria for case confirmation:

At least one of the following five:

1. TBE specific IgM AND IgG antibodies in blood;

2. TBE specific IgM antibodies in CSF;

3. Seroconversion or 4-fold increase in TBE-specific antibodies in paired serum samples;

4. Detection of TBE viral nucleic acid in a clinical specimen;

5. Isolation of TBE virus from clinical specimen.

—Laboratory criteria for a probable case:

Detection of TBE-specific IgM-antibodies in a unique serum sample.

III. Epidemiological Criteria

Exposure to a common source (unpasteurized dairy products).

Case Classification

A. Possible case NA.

B. Probable case: Any person meeting the clinical criteria and the laboratory criteria for a probable case or any person meeting the clinical criteria and with an epidemiological link.

C. Confirmed case.

Any person meeting the clinical and laboratory criteria for case confirmation. 

All of the included patients had confirmed TBE.

In all cases, the diagnosis of TBE was confirmed by detection of specific antibodies with enzyme-linked immunosorbent assay (ELISA) using the kit of Virion/Serion (Wurzburg, Germany) according to manufacturer’s instructions during hospitalization. 

The following case definition of TBE was used. Clinical criteria: a person with (clinical) symptoms of inflammation of the central nervous system (CNS); for example, meningitis, meningoencephalitis, encephalomyelitis, encephaloradiculitis. Meningitis was diagnosed on the basis of inflammatory parameters in CSF with no focal neurological symptoms. Meningoencephalitis was diagnosed when there were inflammatory parameters in CSF, altered consciousness and presence of focal neurological symptoms. Meningoencephalomyelitis was diagnosed when apart from meningoencephalitis symptoms also flaccid paralyses of the limbs were present.

In 40 patients, meningitis was diagnosed; in 26 patients, meningoencephalitis was diagnosed; none of the analyzed patients suffered from meningoencephalomyelitis.

Sequelae were defined as symptoms that persisted or appeared at least one month after the first hospitalization.

In 31 patients (23 males and 8 females, mean age 43.84 ± 12.21 years), sequelae were diagnosed.

All patients were local inhabitants and did not report recent journeys abroad; therefore, there was no need for cross-reactive reaction with other Flaviviridae exclusion. Up to now, no confirmed human case of West Nile virus or yellow fever virus infection was reported in Poland.

TBE antibody titers in serum and CSF samples were measured with an Anti-TBEV ELISA (IgM, IgG) EUROIMMUN test. Serum samples were diluted in 1:404 and CSF 1:2 volume ratio with diluent provided by manufacturer. Intrathecal production of specific immunoglobulins is measured by relative CSF/Serum Quotient (CSQrel), which is the proportion of the quotient of specific immunoglobulins in a particular class (IgM or IgG) and the whole presence of IgM/IgG in CSF to the quotient of specific immunoglobulins in a particular class (IgM or IgG) and the whole presence of IgM/IgG in serum. 

The calculation of the specific antibody index was performed according to manufacturer instructions:

If CSQtotal < CSQlim, the relative CSF/serum quotient is calculated as follows: (1)$$\mathrm{CSQrel}=\frac{\mathrm{CSQspec}}{\mathrm{CSQtotal}}\;$$

If CSQtotal > CSQlim, the CSQlim will be used in the calculation of the relative CSF/serum quotient: (2)$$\mathrm{CSQrel}=\frac{\mathrm{CSQspec}}{\mathrm{CSQlim}}\;$$
(3)$$\mathrm{CSQlim}=0.93\times \sqrt{{\left(\mathrm{CSQalb}\right)}^{2}+6\times 10^{-6}}-1.7\times 10^{-3}\;$$
CSQrel value more than 1.5 means intrathecal specific anti-TBEV antibodies production; 1.3–1.5 values mean borderline results; values 0.6–1.3 mean physiological norm, with no intrathecal antibodies production; and values less than 0.6 mean unreliable results.

To assess the dynamics of intrathecal synthesis, the IgGII/IgGI ratio was calculated (by dividing the IgG index from control examination by the IgG from initial examination).

Statistical analysis was performed using Statistica 12. Groups were compared by Mann–Whitney U test. ROC curve analysis was performed for potential biomarkers. 

## 5. Conclusions

The results of the study suggest the usefulness of the IgG intrathecal index in early diagnosis of TBE sequelae, although further studies are needed. 

## Figures and Tables

**Figure 1 pathogens-11-00416-f001:**
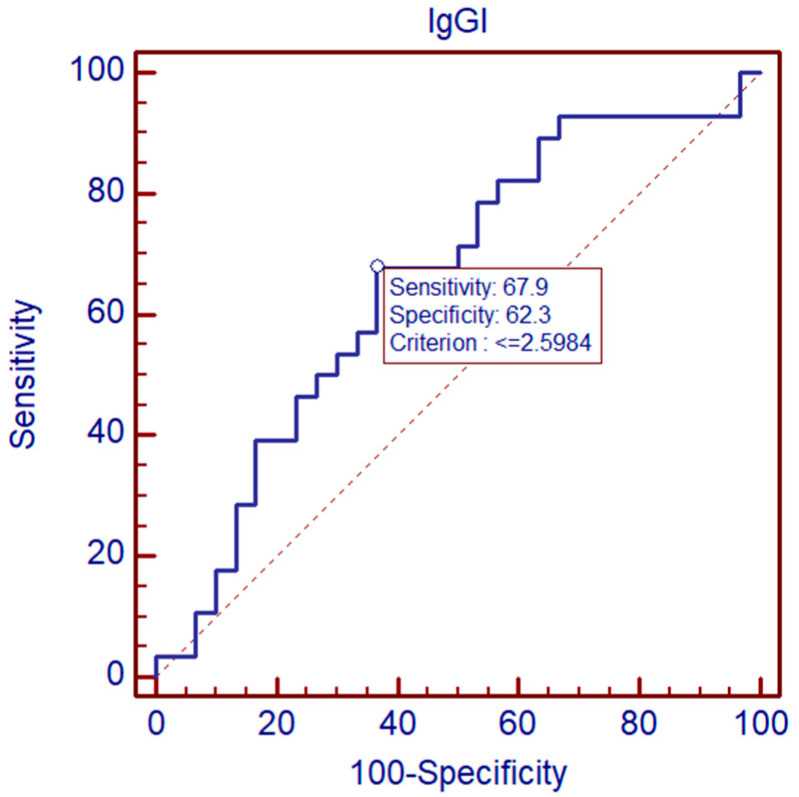
Comparison of IgG1 intrathecal index between sequalae and non-sequelae TBE patients by ROC curve.

**Figure 2 pathogens-11-00416-f002:**
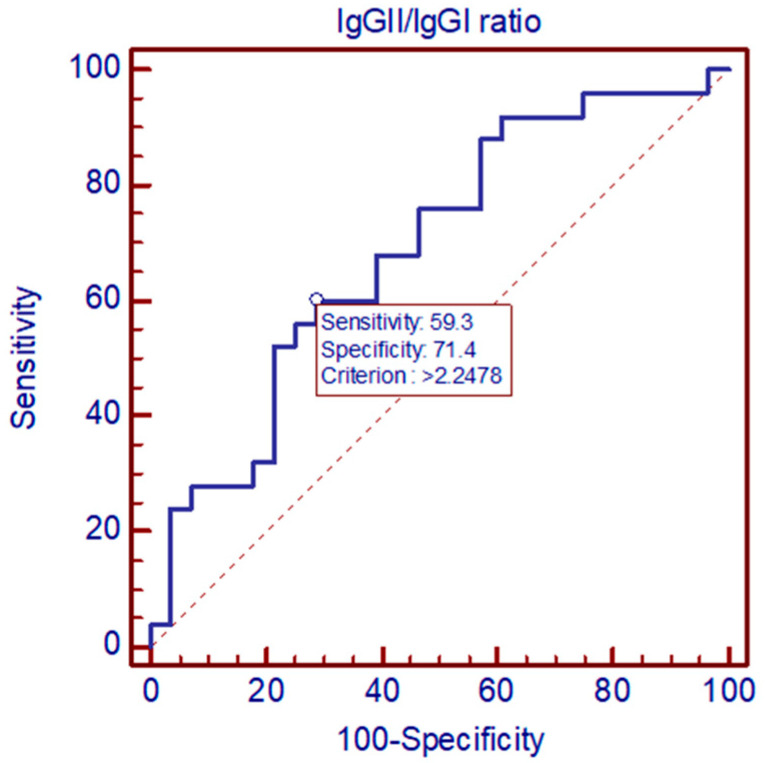
Comparison of IgG2/IgG1 intrathecal index between sequelae and non-sequelae TBE patients by ROC curve.

**Table 1 pathogens-11-00416-t001:** Anti-TBE intrathecal synthesis between male vs. female, meningitis vs. meningoencephalitis, monophasic vs. biphasic co-infection, with *B. burgdorferi*, treatment with dexamethasone, age, sequalae development.

	IgGI				IgGII				IgGII/IgGI			
	Median	Min	Max	*p*	Median	Min	Max	*p*	Median	Min	Max	*p*
**Overall**	2.571	0.4343	12.502		4.035	0.9752	59.679		1.997	0.1466	11.297	
**sex**												
**Male**	2.596	0.4343	12.502	0.59	3.770	1.0405	59.679	0.39	1.997	0.1466	11.297	0.87
**Female**	2.058	0.8051	11.701		4.894	0.9752	25.629		1.722	0.3657	7.666	
**Clinical form**												
**Meningitis**	2.015	0.4343	12.502	0.19	3.775	0.9752	59.679	0.86	2.017	0.3657	11.297	0.41
**Meningoencephalitis**	3.342	0.9125	7.736		4.663	1.0405	32.016		1.997	0.1466	7.798	
**Course**												
**Monophasic**	3.343	0.9152	12.502	** 0.001 **	3.871	1.1153	59.679	0.83	1.363	0.1466	8.055	** 0.04 **
**Biphasic**	2.068	0.4343	12.502		4.585	0.9752	59.679		2.154	0.3657	8.055	
**Coinfection with *B. burgdorferi***												
**No**	2.068	0.4343	12.502	0.12	4.585	0.9752	59.679	0.25	2.154	0.3657	8.055	** 0.02 **
**Yes**	3.11	1.283	8.352		3.627	1.115	14.491		0.792	0.147	11.297	
**Treatment with dexamethasone**												
**No**	2.596	0.4343	12.502	0.53	4.432	0.9752	59.679	0.46	2.01	0.3657	11.297	0.88
**Yes**	2.015	0.9125	8.3		3.514	1.1153	32.016		1.997	0.1466	7.798	
**Age**												
**<60**	2.545	0.4343	12.502	0.52	4.103	0.9752	59.679	0.63	2.069	0.3657	11.297	0.31
**>60**	2.774	1.091	7.606		3.968	1.041	14.935		0.863	0.147	5.591	
**Sequalae**												
**No**	3.05	0.81	12.5	** 0.04 **	3.87	0.98	59.68	0.85	1.27	0.15	8.05	** 0.02 **
**Yes**	1.88	0.43	11.7		4.43	1.68	32.02		3.0	0.37	11.3	

**Table 2 pathogens-11-00416-t002:** Symptoms and sequelae reported by TBE patients.

n = 66		
Symptom	Number of Patients	%
**Symptoms at admission**		
fever	63	95%
fatigue	25	38%
headache	66	100%
nausea	33	50%
paresthesia	3	5%
meningeal signs	53	80%
cerebellar syndrome	18	27%
paresthesia	7	11%
tremor	7	11%
paresis	4	6%
**Persistent sequelae (after 1 month)**		
headache	20	30%
fatigue	7	11%
cerebellar syndrome	2	3%
tremor	2	3%
paresthesia	3	5%
concentration disorders	3	5%

**Table 3 pathogens-11-00416-t003:** Laboratory parameters in CSF and serum at admission and after 1 month.

Parameter	First Examination			Second Examination		
	Median	Min	Max	Median	Min	Max
	CSF					
pleocytosis (1/µL)	93	20	491	17.5	2	116
protein concentration (mg/dL)	67	38	136	47	21	175
albumin concentration [mg/dL]	47.07	23.38	423	29.63	4.01	106.56
TBE IgM [U/mL]	77.9	5.9	2194.7	147.25	0.4	310.7
TBE IgG [U/mL]	620.2	47.3	8500	1786.4	51.2	9194.4
	**Serum**					
CRP [mg/L]	4.18	0	67.53	0.44	0	36.17
WBC [10³/µL]	9.88	4.83	16.02	5.85	3.9	12
PLT [10³/µL]	220	120	675	229	80.6	976
ALT [U/L]	17	5	80	20	1	137
AST [U/L]	17	8	92	19	12	89
TBE IgM [U/mL]	134.2	26.4	1382.3	127.1	2.86	2622.2
TBE IgG [U/mL]	712.55	113.9	8500	1900.9	89.7	7838.5

## Data Availability

Not applicable.

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
