# Peer review of "Anti-TBE Intrathecal Synthesis as a Prediction Marker in TBE Patients"

_pathogens, 2022, doi:10.3390/pathogens11040416_

Round 1

Reviewer 1 Report

Small number of patients - results could be biased. Please add 1) which subtype of TBEV was circulating in the study period, 2) the TBE incidence in Podlaskie voivod and Poland. Please explain the TBE case classification in the study: were the TBE cases probable or confirmed (give confirmed case definition used in EU). What was the vaccination status of the patients? Serological investigation results should be interpreted according to TBEV/or other flavivirus vaccination status; in such cases confirmed cases should be validated by neutralization/or equivalent assay.

Reviewer 2 Report

The claim: "IgG index might be an useful tool in prediction of TBE sequelae develompent" has not beyond doubt shown. he U-test p 0.02 is irrelevant considering the small sample size with overlappin ranges of i.t. specific antibody indices. The ROC test p<0.05 is not a measure of the predictive performance, cf. positive likelihood ratio 1.8 giving a probability approximation of 15%. The predictive perdomance should be more thoroughly discussed.

The presentation of athe calculation of specific antobody index could be mor clear by including the formula. Te relevance of BBB damage as reflected by QAlb for the index calculation should be considered.

Round 2

Reviewer 2 Report

Revison performed as suggested, thank you.